# A Systematic Review on the Emergence of Omicron Variant and Recent Advancement in Therapies

**DOI:** 10.3390/vaccines10091468

**Published:** 2022-09-05

**Authors:** Beyau M. Konyak, Mohan Sharma, Shabnam Kharia, Ramendra Pati Pandey, Chung-Ming Chang

**Affiliations:** 1Integrated Molecular Diagnostic and Research Laboratory (BSL-2), District Hospital Tuensang, Tuensang 798612, Nagaland, India; 2Department of Life Sciences, Dibrugarh University, Dibrugarh 786004, Assam, India; 3Centre for Drug Design Discovery and Development (C4D), Department of Biotechnology & Microbiology SRM University, Delhi-NCR, Rajiv Gandhi Education City, Sonepat 131029, Haryana, India; 4Master & Ph.D. Program in Biotechnology Industry, Division of Biotechnology, Chang Gung University, No. 259, Wenhua 1st Rd., Guishan Dist., Taoyuan City 33302, Taiwan

**Keywords:** emergence of Omicron and its mechanism, mutation and sub-lineages, monoclonal antibodies, anti-viral drugs

## Abstract

With the ongoing COVID-19 pandemic, the emergence of the novel Omicron variant in November 2021 has created chaos around the world. Despite mass vaccination, Omicron has spread rapidly, raising concerns around the globe. The Omicron variant has a vast array of mutations, as compared to another variant of concern, with a total of 50 mutations, 30 of which are present on its spike protein alone. These mutations have led to immune escape and more transmissibility compared to other variants, including the Delta variant. A cluster of mutations (H655Y, N679K, and P681H) present in the Omicron spike protein could aid in transmission. Currently, no virus-specific data are available to predict the efficacy of the anti-viral and mAbs drugs. However, two monoclonal antibody drugs, Sotrovimab and Evusheld, are authorized for emergency use in COVID-19 patients. This virus is not fading away soon. The easiest solution and least expensive measure to fight against this pandemic are to follow the appropriate COVID-19 protocols. There is a need to strengthen the level of research for the development of potential vaccines and anti-viral drugs. It is also important to monitor and expand the genomic surveillance to keep track of the emergence of new variants, thus avoiding the spread of new diseases worldwide. This article highlights the emergence of the new SARS-CoV-2 variant of concern, Omicron (B.1.1.529), and the vast number of mutations in its protein. In addition, recent advancements in drugs approved by FDA to treat COVID patients have been listed and focused in this paper.

## 1. Introduction

The emergence of this ongoing pandemic coronavirus has devastated the world. It has been almost two years since COVID-19 was detected in Wuhan, China. As of 12 August 2022, more than 6 million deaths and 585 million confirmed cases have been reported worldwide, making it one of the deadliest viruses in history [1]. This virus has undergone several recombination and mutations over the past years, leading to the emergence of a vast array of new variants of concern (VOC) and variants of interest (VOI), resulting in a high level of global health alerts and panic.

While the world has been tackling the Delta variant, the emergence of the novel Omicron variant in December 2021 took the world by storm [2]. The first SARS-CoV-2 variants to emerge were Alpha, Beta, Gamma, and Delta, and the Omicron (B.1.1.529) variant, classified as VOC by WHO [3] on 26 November 2021, has become the dominant one in many countries since January 2021 (Table 1). The novel Omicron variant has a larger number of mutations as compared to other VOCs. This has led to a higher rate of infections, higher transmissions, and immune escape from COVID-19 vaccines, resulting in rapid spread worldwide in a shorter period. As per the WHO, this novel variant also enhances immune or diagnostic evasion, causes less severe disease, and shows enhanced transmissibility. In addition, it poses a detrimental impact on epidemiology and shows increased variability in clinical presentation. The four most common symptoms of an Omicron infection, according to the US Centers for Disease Control and Prevention, are cough, fatigue, congestion, and a runny nose [4]. Although the Omicron variant seems more contagious than previous variants, as per the data available, early findings suggest lower rates of hospitalization for Omicron-positive patients, as compared to the Delta variant. However, the Omicron variant should not be taken lightly. Precautionary measures should be taken such as avoiding crowded spaces, maintaining social distances, and wearing masks to prevent the spread of infection [5]. The SARS-CoV-2 Interagency Group (SIG) evaluates and classifies the novel variant as a VOC, based on the cases detected in various countries, its travel history, rate of transmission, mutations in the spike protein, and as per data available for other variants. It is also evaluated based on a reduction in the effectiveness of vaccines and monoclonal antibody treatments. The SIG tracks and evaluates the emergence of new viruses and actively monitors the potential impact of vaccines, therapeutics, and diagnostics against SARS-CoV-2 [6]. This review article highlights the emergence of the Omicron variant, its mechanism, and the vast array of mutations in the spike protein, which enhance transmissibility and immune escape from vaccines. The potential impact on diagnosis, the various advancements in anti-viral drugs, and therapeutic treatments to tackle the ongoing COVID-19 pandemic is also discussed.

## 2. The Emergence of the Omicron (B.1.1.529) Variant and Its Sub-Lineage

South Africa was the first country to detect the novel variant of SARS-CoV-2 from specimens collected on 11 November and 14 November 2021. The first Omicron variant was reported to WHO by the South African Department of Health on 24 November 2021 [6]. On 25 November 2021, the United Kingdom Health Security Agency designated the B.1.1.529 Omicron variant as a Variant Under Monitoring (VUI-21-NOV-01) and, just one day later, it was designated as a VOC by the WHO; the variant was later named Omicron [4,7]. The rapid spread of this variant in South Africa and surrounding countries alarmed the WHO. This could be due to the large number of mutations on the spike protein in the Omicron variant as compared to other VOCs. However, Omicron has been reported to be less severe than the previous variants [8,9]. Three days after the announcement of Omicron on 29 November 2021, by the WHO, this novel variant was detected in Australia, Belgium, Canada, the Czech Republic, Denmark, France, Germany, Italy, Netherlands, and the United Kingdom [10]. As of 25 December 2021, Omicron had been reported in 108 countries, with nearly 150,000 confirmed cases and 26 deaths, creating an alarming situation worldwide [11]. The emergence of the Omicron variant in South Africa has also revealed a vaccination gap between the richest and the poorest. Whilst an average of 60% of the population of Europeans has been vaccinated, it is appalling to see that only 5–10% of the African population has been vaccinated [12]. This could be one of the major factors contributing to the emergence of VOCs. Japan also reported the first two cases of the novel Omicron variant at the end of November 2021 among international travelers returning to the country with the undetected Omicron variant [13]. India reported the first two cases of the Omicron variant on 3rd December 2021 among international travelers [14]. In Germany, two suspected cases of the new Omicron variant were first reported on 27 November 2021 and, as the New Year began, this novel Omicron became one of the most dominant variants in Germany, despite protective measures at the initial stage in January 2022 [15,16]. This new variant has been spreading worldwide and has been reported in 180 countries as of 4 February 2022. As of 11 February 2022, the UK reported a maximum number of Omicron cases (422,993) followed by the USA (359,236), Denmark (69,789), Germany (48,866), Canada (30,687), and France (27,487), while an increasing number of cases can be seen on a daily basis in many other countries [16]. Jansen et al. [17] reported that the Omicron variant has a shorter incubation period than the previous variants, with similar clinical symptoms. Preliminary data conducted by the National Institute of Infectious Diseases shows a high viral load three to six days after the onset of symptoms [18].

As the Omicron variant continues to evolve and mutate, new sequences (sub-lineages) of the Omicron variant were reported by the WHO in December 2021, namely: BA.1, BA.2, and BA.3 [19]. Over the past two months, the BA.1 variant has escalated globally. However, in the last weeks of January 2022, the BA.2 sub-lineage had increased internationally and had already become dominant in Denmark. As of 11 February 2022, BA.2 had been detected in at least 70 countries and 44 U.S. states, according to the data uploaded in GISAID, and a total number of 55,404 sequences in the BA.2 lineage have been detected since the lineage was identified [20,21]. A preprint uploaded in medRxix [22], concluded that Omicron BA.2 is more dominant and transmissible than the parent Omicron variant (BA.1). It possesses immune-escaped advantages among vaccinated individuals; however, it does not increase its transmissibility to vaccinated individuals with breakthrough infections. India is another country where BA.2 is rapidly replacing the Delta and Omicron BA.1 variants, as per the report [23]. Data uploaded on GISAID suggest that the BA.2 variant is increasing rapidly in proportion to the original BA.1 variant, thus dominating the parental variant based on the sequence frequency [24].

## 3. Mechanism of Omicron Variant

The Omicron variant has the highest number of mutations observed so far, as compared to the other VOCs such as Alpha, Beta, Gamma, and Delta. This variant contains more than 30 mutations in the spike protein, which binds to the ACE2 receptor in the human cell for invasion, and out of which, 15 modifications are observed in a Receptor Binding Domain (RBD), an area strongly associated with humoral immune evasion [25], thus, increasing its transmissibility and thereby causing the possibility of evading vaccine-induced antibodies [26]. Researchers from China constructed and studied the binding affinity between the complex structure of the human ACE2 protein and the RBD of the Omicron variant spike protein (S-protein) using atomistic molecular dynamics simulations. They observed that mutations in the RBD of the SARS-CoV-2 Omicron variant resulted in a higher binding affinity to the human ACE2 protein [27].

There are two distinct mechanisms for the entry of SARS-CoV-2 into a human cell: cell-surface entry and endosomal entry, where TMPRSS2-mediated S protein activation takes place at the plasma membrane and cathepsin-mediated activation takes place in the endolysosome at the cytoplasmic region, respectively [28]. A researcher from Glasgow University in the UK demonstrated the replication process of different SARS-CoV-2 variants and their mechanisms in entering the human cell. An HIV-pseudotype was evaluated for entry assay. The study reveals that Omicron, like pangolin CoV, uses the endocytosis pathway to enter human cells and is independent of the TMPRSS2 present on the cell surface. This study was further supported by using drug inhibitors targeting either the TMPRSS2 (Camostat) or the cathepsins (E64d). As compared to another variant spike, the Omicron spike shows a reduced syncytium formation, thereby causing fewer lung infections [29].

## 4. Mutations in Omicron Variant

In November 2021, Omicron (B.1.1.529) emerged as a VOC. Wahid et al. [26] had earlier reported the triplet mutation (K417N + E484K + N501Y) in the Beta (B.1.351) and Gamma (P.1) variants, which were highly transmissible, leading to greater COVID-19 hospitalizations, ICU admissions, and even deaths. Such mutations have increased the immune escape potential leading to a decrease in the neutralization of antibodies (such as those found in the Pfizer and Moderna vaccines). The Omicron variant contains overall 50 mutations, with 30 mutations [25] on the spike protein, which include: A67V, del69/70, T95I, G142D, del143/145, N211I, del212/212, G339D, S371L, S373P, S375F, S477N, T478K, E484A, Q493R, G496S, Q498R, N501Y, Y505H, T547K, D614G, H655Y, N679K, P681H, N764K, D796Y, N856K, Q954H, N969K, and L981F, of which 15 modifications are located at the RBD region [30]. Cecylia et al. [27] conducted atomistic molecular dynamics simulations to study the binding interactions between the human ACE2 protein and the RBD of the Omicron spike protein (S-protein). The analysis shows that the Omicron RBD binds more strongly to the human ACE2 protein than does the original Wuhan strain. However, the SARS-CoV-2 Omicron RBD shows weaker binding affinity than the Delta variant [31]. Most COVID-19 strains contain at least one change from the original Wuhan sequence, D614G, altering the virus’s ability to escape the immune response. The N501Y mutation is present in all VOCs, except the Delta variant [32]. Interestingly, Omicron’s spike protein mutations, such as D614G, N501Y, and K417N, are found in some other VOCs, thus making the virus more infectious—a prospect that is very concerning. 

Similarly, the H655Y, N679K, and P681H mutations in the Omicron spike protein, also found in the Alpha and Delta variants, could increase the transmission of the virus [6,33,34]. It has the deletion at spike protein (Δ69–70), position 69–70, similar to the Alpha and Eta variants, that leads to the S-gene dropout or S-gene target failure. This phenomenon may provide a useful proxy to measure the prevalence of the Omicron variant. The rapid transmission of the Omicron variant across the globe could be due to the presence of these larger numbers of mutations in the spike protein, unlike the Delta variant, and thus evading immune response [33]. However, this phenomenon does not apply to the BA.2 sub-lineage of the Omicron variant, as it does not contain the deletion at S: 69–70 and is S-gene target positive (SGTP) on PCR diagnostic assays [35]. The comparison and mutual sharing of the spike protein mutations of the Omicron sub-lineages, BA.1, BA.2, and BA.3, are represented by a Venn diagram, as shown in Figure 1.

The combination of N501Y + Q498R may increase the binding affinity to ACE2, while other substitutions may lead to a decrease in the binding affinity to the ACE2 in the Omicron spike protein. Kinases’, such as PI3K/AKT, signaling are essential in SARS-CoV-2 entry. Based on molecular docking, Bexultan et al. [36] analyzed the interaction between the potential kinases and the N501Y mutation and observed that the N501Y mutation did not enhance binding to epidermal growth factor receptors (EGFR) due to the mutations. The N501Y mutation-containing lineages might become more infectious since several kinases are elevated in cancer patients. Therefore, additional care for cancer management should be taken into consideration. Mannar et al. [37] analyzed a cryo-electron microscopy structure between the spike proteins of the Omicron variant in complex with human ACE2. The structure depicts two additional new salt bridges and hydrogen bonds formed by the mutated residues R493, S496, and R498 in the RBD with the ACE2. This enhances other mutations, such as K417N, which is known to reduce the ACE2 binding affinity to the spike protein. These strong interactions at the ACE2 interface may contribute to the rapid spread of the Omicron variant [38]. Zhuoming et al. [39] reported that E484K could escape neutralization by convalescent sera, while S477N was resistant to neutralization by multiple mAbs, thus needing further investigation. The increase in binding affinity to the ACE2 receptor by N501Y could aid in an increase in transmission. The combination of N501Y and Q498R may also increase the binding affinity even more; however, other substitutions in the Omicron spike protein are expected to decrease binding to the ACE2. A cluster of mutations (such as H655Y, N679K, P681H) present at the S1–S2 furin cleavage site in the Omicron variant may increase spike cleavage and could aid in transmission. The N679K present at the furin cleavage site adds to the polybasic nature and is associated with an increase in transmission. The P681H mutation (also found in Alpha) enhances spike cleavage, and could also aid in transmission. At this position, an alternate mutation (P681R) is found in Delta [40]. In India, Surendra et al. [41] found a major mutation at the P681 position with an R (P681R) similar to the Delta variant (B.167.2) of about 9.71% instead of the H (P681H) mutation. This variant is of considerable public concern, as it is increasing at a high rate in many countries, including the U.S., due to its increased transmissibility and immune evasion. In the nucleocapsid (N) protein region, the Omicron variant also possesses two additional mutations such as R203K and G204R (found in ancestral mutation) that enhance sub-genomic RNA expression and viral RNA binding with key host proteins, thus increasing the viral load [42,43]. Phylogenetic analysis, based on the prevalence of high numbers of mutations, revealed that the Omicron variant is closely related to the Gamma (P.1) variant [44], and may possess similar characteristics at the molecular level [45].

## 5. Impact of Diagnosis on the Omicron Variant

The emergence of mutations in SARS-CoV-2 has resulted in five SARS-CoV-2 variants, namely: the Alpha, Beta, Gamma, Delta, and Omicron variants. The World Health Organization has designated such variants as VOC [46]. This has resulted in investigating the performance of the potential impact on diagnosis. A rapid diagnostic antigen test is cheap and offers quick results at the point of care when the viral load is high and, hence, provides utility in clinical and public health settings. The rapid test is approved in many countries, including Australia, for self-testing [47]. However, it is less sensitive than the RT-PCR-based method [48].

Deerain et al. [49] evaluated ten commercially available rapid antigen test kits, based on nucleocapsid protein antigens, to check their diagnostic performance (sensitivity) between the Delta and Omicron variants. Overall, they observed no sensitivity difference among the variants. The detection limit of all the kits for the Delta variant was 6.50 log10 copies per mL (Ct 25.4), and 6.39 log10 copies per mL (Ct 25.8) for the Omicron variant. Similarly, Puhach et al. [50] evaluated seven Ag-RDTs for their sensitivity to the Omicron variant and compared them to other VOCs such as Alpha, Beta, Gamma, and Delta. A trend toward lower sensitivity for Omicron detection compared to the other VOCs was observed. As per the data retrieved on 28 December 2021 (the Food and Drug Administration), it suggests that the antigen test provides a rapid result, and does detect the Omicron variant, but is not a very reliable or sensitive diagnostic testing option [51].

The Paul Ehrlich Institute (PEI) has also investigated and evaluated the sensitivity of 198 rapid antigen tests (Qualitative Lateral Flow Tests) using uniform sample material marketed in Germany until the end of January 2022. The current state-of-the-art was defined as corresponding to a minimum sensitivity of 75% for the pools with Cq ≤ 25. However, the results do not allow any conclusions regarding the specificity of the tests [52]. Some studies have also demonstrated that up to 50% of the positive cases detected by rapid tests are false positives as compared to PCR diagnostics tests. A recent small preprint study that has not been peer-reviewed found that the antigen test fails to detect the virus on day zero of 30 individuals who turn out to be COVID positive in the PCR test. The PCR tests also indicated a higher viral load [53]. However, a study performed in January 2022, in California, on 731 individuals’ shows that the Abbott BinaxNOW rapid tests could detect the Omicron variant, as they did with other variants, especially when people have higher viral load and were symptomatic [54]. The Abbott BinaxNOW Rapid Antigen Test can be a useful adjunct to RT-PCR testing for the detection of SARS-CoV-2 infections [55]. 

The RT-PCR test is considered to be the ‘gold standard’ for detecting coronavirus. In the RT-PCR, two or more spike genes are targeted; therefore, there are chances to detect one of the genes in case of any mutations, thus highlighting its advantage for the Omicron variant. Ever since the detection of SARS-CoV-2 in 2019, it has undergone a broad process of replication and mutation in the spike proteins and RBD, generating a vast number of VOIs and VOCs. Due to this mutation, a phenomenon called ‘S-gene dropout’ or ‘S-gene target failure’ has been reported [7]. This vast mutation has jeopardized the reliability of currently used diagnostic kits for detecting SARS-CoV-2 and, hence, requires a regular re-evaluation of commercially available kits based on the emerging VOIs and VOCs [56]. ‘S-gene target failure’ has been suggested as a marker in identifying the Omicron variant. However, with the emergence of the new BA.2 sub-lineage of Omicron (which does not contain 69/70del in the spike protein), it cannot be considered as a marker for the presence of Omicron and requires confirmation of Omicron by sequencing [4]. This phenomenon has been supported by Metzger et al. [57], who obtained a piece of information and collected 39 assays of the most commonly used PCR tests in Switzerland and Liechtenstein, targeting genomic loci such as the ORF1ab region, the RdRp gene, the S gene, the E gene, the N gene, and the M gene. Only two assays showed S-gene dropout for Omicron out of eight assays targeting the S gene. Hence, gene sequencing data analysis is required to confirm the presence of the Omicron variant.

Whole-genome sequencing of Omicron with next-generation sequencing (NGS) might serve as a gold standard to detect SARS-CoV-2 variants despite being time-consuming and costly. It also requires large data processing. Fu et al. [58] compared and evaluated the Allplex SARS-CoV-2 Master Assay and Variants I Assay (a direct PCR-based variant analysis) to detect HV69/70 deletion, Y144 deletion, E484K, N501Y, and P681H spike mutations with NGS for 115 samples and observed sensitivity of 98.7% with the Master Assay and 100% with the Variants I Assay. These assays can be utilized as useful tools to rapidly monitor selected and updated VOCs in resource-limited settings. Currently, the CDC is working to understand the new Omicron variant and the effectiveness of commercially available diagnostic tools and authorized medical countermeasures, such as vaccines and therapeutics, against this variant, and is providing technical support to monitor the epidemiologic and clinical features of novel variants [59].

## 6. Advancement in Therapeutics Drugs

The emergence of COVID-19 has led to the development of various repurposed and therapeutic drugs, including antivirals and antibody drugs. The U.S. Food and Drug Administration (FDA) have authorized the use of these drugs for emergency purposes in COVID-19 patients with serious illnesses, as per the panel’s recommendation. Meanwhile many anti-viral drugs and mAbs are under investigation.

The in vitro studies performed by IGM Biosciences, Inc. indicate the novel antibody IGM-6268 exhibits potential neutralizing activity against the Omicron variant and all other VOCs and VOIs [60]. Some of the recent advances in therapies that have been approved and authorized for emergency use in COVID-19 patients and are suitable for the Omicron variant are being discussed in this section (Table 2).

### 6.1. Immunomodulatory Drugs

Amid Omicron, WHO continues to approve new COVID-19 treatments for hospitalized patients, according to their disease severity. The arthritis drugs tocilizumab and sarilumab [61,62], which have been found to have efficacy in the treatment of COVID-19 patients with moderate to severe COVID-19 pneumonia, have recently been approved by WHO [63]. Similarly, dexamethasone, which is effective at reining in lung-damaging inflammation [64], and Baricitinib a Janus kinase (JAK) inhibitor, are also approved by WHO for treatment in COVID-19 patients as per the panel’s recommendation [65]. Recently, studies have shown that dextromethorphan treatment was characterized by regulation of adaptive immunity and other specific local innate; however, it was not associated with the regulation of pro-inflammatory pathways in COVID-19 acute respiratory distress syndrome (CARDS) [66].

### 6.2. Antivirals Therapy

Recently, the U.S. FDA issued an emergency use authorization (EUA) for Pfizer’s Paxlovid (nirmatrelvir tablets and ritonavir tablets) for the treatment of COVID-19 patients with mild–moderate symptoms and those at high risk of progressing into severe illness. This should be prescribed immediately, within five days after diagnosis of COVID-19 [67]. During the biochemical assay, the nirmatrelvir drug was shown to inhibit the 3CL protease associated with the Omicron (B.1.1.529) variant [68]. In vitro studies also suggest that Paxlovid has the potential to maintain plasma concentrations, thus preventing Omicron and other variants from replicating.

Molnupiravir, an oral antiviral drug, is a small molecule of the synthetic nucleoside derivative N-hydroxycytidine (NHC). It targets viral RNA polymerase and inhibits SARS-CoV-2 replication. It was granted EUA by the FDA’s Antimicrobial Drugs Advisory Committee on 23 December 2021, for the treatment of COVID-19 [69].

The antiviral drug Veklury (remdesivir) was approved by the FDA in 2020 for the treatment of COVID-19 in adult and pediatric patients, and those of old age [70]. These three drugs: molnupiravir, remdesivir, and paxlovid have shown neutralizing activity against other VOCs, including the novel Omicron variant [71]. Many antiviral drugs such as Ivermectin, Interferon Alfa, Interferon Beta, Interferon Lambda, and Nitazoxanide are still under evaluation for the treatment of COVID-19. All these drugs have paved the way as new hopes to fight against the COVID-19 pandemic.

### 6.3. Monoclonal Antibodies

The emergence of COVID-19 has led to the development of various treatments. Monoclonal antibodies (mAbs) are a novel class of antiviral intervention [72]; they target the RBD of the SARS-CoV-2 spike protein, which is highly mutated in the Omicron variant [73]. They are one of the most effective therapeutic strategies to neutralize viral replication. This has been shown proven in a previous experiment conducted by researchers from the Baylor University Medical Centre, Dallas, Texas in the US, where dual mAb therapy was shown to reduce viral transmission [74].

Currently, four anti-SARS-CoV-2 mAb products: bamlanivimab plus etesevimab, casirivimab plus imdevimab (REGEN-COV), sotrovimab, and tixagevimab plus cilgavimab (Evusheld) have been authorized by the FDA for emergency used to treat COVID-19-positive, non-hospitalized patients who are likely to develop more serious disease at the later stage [75]. The bebtelovimab monoclonal antibody was approved recently [76]. Such mAbs as casirivimab plus imdevimab, and bamlanivimab plus etesevimab, which have shown effectiveness in the previous VOCs, have reduced neutralization in the Omicron variant [77].

Recently, Emi et al. [78] assessed the neutralizing activities of mAbs using a live-virus focus reduction neutralization assay (FRNT) against the Omicron variant and other VOCs. Etesevimab, bamlanivimab and imdevimab did not neutralize Omicron. Casirivimab showed reduced neutralization against the Omicron variant. COV2-2196 (tixagevimab), COV2-2130 (cilgavimab), and S309 (marketed as sotrovimab) retained neutralizing activity against the Omicron variant. Thus, there are two authorized monoclonal antibody treatments against Omicron—Sotrovimab and Evusheld. Early reports also suggest that bamlanivimab and C144-LS antibodies have reduced efficacy against the Omicron variant. Based on early modeling, studies have shown casirivimab plus imdevimab (REGN-COV2), as well as the Rockefeller University antibody C135, to be effective against the Omicron variant [79]. Some of the monoclonal antibodies authorized by the FDA for emergency use are listed below.

#### 6.3.1. Bamlanivimab and Etesevimab

These mAbs bind at the overlapping sites in the spike protein RBD of SARS-CoV-2; blocking its attachment to the human ACE2 receptor. The FDA has banned the use of these mAbs in some areas of the United State due to the progression of variant resistance [80]. However, the FDA has approved a EUA to allow the use of this mAbs injection in certain non-hospitalized adults and children and infants (≥2 years of age) who have mild to moderate COVID-19 symptoms with a risk of progressing to serious illness [81].

#### 6.3.2. Casirivimab and Imdevimab mAb

These are recombinant human mAbs that prevent the entry of the virus into human cell by targeting the spike protein of SARS-CoV-2. The FDA has authorized the use of these mAbs to treat mild to moderate SARS-CoV-2-positive adults and pediatric patients. It is also administered to patients above 65 years of age with chronic medical conditions. However, it is not authorized for hospitalized COVID-19 patients with oxygen therapy [82]. Recently, the FDA rescinded its authorization of these drugs, due to the prevalence of the Omicron variant [83].

#### 6.3.3. Sotrovimab

This mAb targets an epitope region in the spike protein RBD that is conserved between SARS-CoV and SARS-CoV-2. This monoclonal antibody has been shown to neutralize the Omicron variant. However, it is not authorized for use in COVID-19 patients who are hospitalized with mechanical ventilation [84].

#### 6.3.4. Tixagevimab Plus Cilgavimab

The tixagevimab and cilgavimab monoclonal antibodies bind to non-overlapping sites of the SARS-CoV-2 spike protein, preventing the interaction between the virus and the ACE2 receptor. The FDA has approved a EUA to allow certain adults and children 12 years of age and older to receive this mAbs injection. The combination has shown to somewhat decrease neutralizing activity in in vitro against the Omicron variant [85].

#### 6.3.5. Bebtelovimab (LY-CoV1404)

This mAbs has not been approved and has only been authorized for emergency use by the FDA for treatment of mild to moderate COVID-19 patients (≥12 years old) with a high risk of progressing to severe illness. It neutralizes the SARS-CoV-2 spike glycoprotein RBD, thereby inhibiting the entry of the virus into a human cell. Bebtelovimab is active and has shown some neutralizing activity against other VOCs, including the Omicron variant [76].

Thus, the abilities of all these monoclonal antibodies and anti-viral therapies to fight and neutralize the Alpha, Beta, Gamma, and Delta VOCs—including the novel Omicron variant—have paved the way for hope in the future to make a potential therapeutic and diagnostic intervention.

**Table 2 vaccines-10-01468-t002:** List of drugs which has been approved and authorized by FDA for emergency use in COVID-19 patients. (Source: FDA homepage, Medlineplus, Medicalletter).

Drugs	Dosage	Method of Administration	Duration	Side Effect	Mode of Action	Efficacy of Monoclonal Antibodies/ Drugs against SARS-CoV-2 Variants	Reference
Sarilumab	Single dose of sarilumab 400 mg	Intravenous infusion	Sarilumab infusion should be used within 4 h of preparation	Neutropenia, thrombocytopenia and GI perforation	Suppress cytokine storm	Effective against COVID-19	[61,63,65]
Tocilizumab (Actemra)	12 mg/kg in patients weighing <30 kg and 8 mg/kg in those weighing ≥30 kg (max dose 800 mg)	Intravenous infusion	Administered as a single IV infusion over 60 min	Neutropenia, thrombocytopenia, serum hepatic transaminase elevations, gastrointestinal perforations, and anaphylaxis	Inhibits binding of the proinflammatory cytokine IL-6 to its receptors	Effective against COVID-19	[61,65]
Dexamethasone (DEX)	DEX 6 mg	Orally or Intravenous infusion	Once daily for up to 10 days or until hospital discharge, whichever comes first	Avascular necrosis, Adrenal insufficiency, Increased BP, Peripheral edema, and Myopathy	Suppress inflammation	Effective against COVID-19	[65]
Baricitinib (Olumiant)	4 mg	Orally	Once daily while hospitalized for up to 14 days	Lymphoma and other malignancies, Thrombosis, GI perforation and serious cardiac-related events (e.g., MI, stroke)	Inhibits viral endocytosis	Effective against COVID-19	[65]
Remdesivir (Veklury)	200 mg IV on day 1, followed by 100 mg once daily	Intravenous infusion	Treatment duration is 5–10 days	Nausea, Hypersensitivity reactions	Inhibits viral RNA-dependent RNA polymerase	Active against Omicron and all other variants: Alpha, Beta, Gamma and Delta	[70,71]
Molnupiravir	800 mg every 12 h	Orally	Treatment should be started within 5 days of symptom onset	Diarrhea, nausea, and dizziness	Targets viral RNA polymerase, inducing mutagenesis and inhibiting SARS-CoV-2 replication	Active against the entire variants: Alpha, Beta, Gamma, Delta and Omicron	[69,71]
Paxlovid (Ritonavir-Boosted Nirmatrelvir)	300/100 mg (2 nirmatrelvir tablets and 1 ritonavir tablet taken together) twice daily	Orally	Treatment should be started within 5 days of symptom onset	Dysgeusia, diarrhea, hypertension, and myalgia	Inhibits the SARS-CoV-2 main protease (Mpro), preventing viral replication.	Active against entire SARS-CoV-2 variant including, omicron	[71]
Bamlanivimab Plus etesevimab	700 mg (one vial) of Bamlanivimab and 1400 mg (two vials) of etesevimab	single Intravenous infusion	One-time dose right after COVID-positive test and within 10 days after the onset of infection symptoms	Bleeding, bruising, pain, soreness, or swelling at injection sit	Binds to spike protein RBD of SARS-CoV-2	Active against two SARS-CoV-2 variant: Alpha and Delta	[76,77,81]
Casirivimab plus imdevimab (REGEN-COV)	600 mg of casirivimab and 600 mg of imdevimabis diluted together in 50, 100, 150, or 250 mL of normal saline at a maximum rate of 310 mL/hr (180 mL/hr if diluted in 50 mL)	Intravenous infusion or Subcutaneous injection	After a positive SARS-CoV-2 test result and within 10 days of COVID-19 symptom onset	Pain, bleeding, bruising of the skin, soreness, swelling, or infection at injection site	Bind at different sites on the RBD of the spike protein of SARS-CoV-2.	Active against all SARS-CoV-2 variant: Alpha, Beta, Gamma, Delta, Omicron	[77,79,82]
Sotrovimab	500 mg of sotrovimab diluted in 50 or 100 mL of normal saline	Intravenous infusion	After a positive viral test for SARS-CoV-2 and within 10 days of symptom onset	Rash (2%) and diarrhea (1%). Hypersensitivity reactions, including anaphylaxis	Binds at the epitope sites on the spike protein and prevent membrane fusion after the virus binds to the human ACE2 receptor	Active against the entire variants: Alpha, Beta, Gamma, Delta and Omicron	[75,78,84]
Tixagevimab plus cilgavimab	150 mg of each antibody is administered.	Intramuscular injection	one time as two shots (one after another)	Headache (6%) and fatigue (4%).	Binds to non-overlapping region of the SARS-CoV-2 spike protein.	Active against the entire variants: Alpha, Beta, Gamma, Delta and Omicron	[75,78,85]
Bebtelovimab	175 mg/2 mL vials	Intravenous infusion	Treatment should be started within 7 days of symptom onset	Rash, pruritus, and infusion-related reactions	Binds to the spike protein and prevents its attachment to the human receptor	Active against the Omicron variant of SARS-CoV-2 (Neutralize BA.1 and BA.2omicron variant)	[76]

## 7. Discussion

The emergence of the novel Omicron variant has created chaos worldwide. Just after the announcement of Omicron as a VOC by WHO, this novel variant had already spread globally. This rapid spread of the Omicron variant has concerned global health bodies. This is due to a high level of mutation; overall, Omicron has 50 mutations, with 30 mutations on its spike protein. This review gives an overview, highlighting concerns about the emergence of the Omicron variant, its vast array of mutations, the impact of diagnosis methods to detect the novel Omicron variant, and the novel advancement in therapeutic drugs.

The Omicron variant possesses a phenomenon called S-gene dropout, due to the deletion of the amino acid at spike protein position 69–70. This acts as a marker to measure the prevalence of the Omicron variant during diagnosis. However, this phenomenon does not apply to the BA.2 sub-lineage of the Omicron variant. A vast number of mutation and phylogenetic analyses revealed that the Omicron variant may possess similar characteristics to the Gamma (P.1) variant at the molecular level, predicting that drugs suitable for gamma may equally work for the novel Omicron variant.

Recent in vitro studies indicate that Novel Antibody IGM-6268 exhibits potential neutralizing activity against the Omicron variant, as well as other VOCs and VOIs. Interestingly, three drugs namely: molnupiravir, remdesivir, and paxlovid have been approved and authorized by the FDA for emergency use, as they show neutralizing activity against other VOCs, including the novel Omicron variant. This article has a limitation, as it lacks clinical data on the efficacy of mAbs and anti-viral drugs for the treatment of patients with the Omicron variant. Here, we have only described mAbs and anti-viral drugs authorized by the FDA for emergency use in hospitalized and non-hospitalized COVID-19 patients.

It is almost inescapable that there will be the emergence of a new variant, as long as there is a loophole in one part of the world that is unprotected.

## 8. Conclusions and Prospect

It has been more than two years since the emergence of this COVID-19 pandemic. Despite several measures and protocols to control the virus, it has dominated the world. High levels of effort and mass vaccination campaigns to restrain the spread of this virus have become useless with the emergence of novel variants. COVID-19 has escalated around the globe within a short period and the outcome is devastating, with global health concerns and economic fallout. The emergence of a new VOC every year with the accumulation of a large number of mutations is of concern. The vast array of mutations in SARS-CoV-2 has recently given rise to the most mutated Omicron variant of SARS-CoV-2. The emergence of mutations could be due to the transmissibility of the virus from person to person, leading to its escape from population immunity and frequent transmission. It is the nature of the virus to mutate. This novel Omicron variant has shown itself to be resistant to vaccination and requires booster doses. The hypothesis of its epizootic transfer needs validation and should be thoroughly investigated. At present, there is no specific data and evidence to get rid of SARS-CoV-2. Moreover, the etiology of SARS-CoV-2 is poorly understood. As of now, some anti-viral and monoclonal antibody drugs are being authorized by the FDA for emergency use, as per the panel’s recommendation. This COVID-19 is not going away soon, and may remain forever. The only way to tackle this situation is to follow appropriate protocol and measurements as per the WHO recommendations, such as quarantine, keeping good personal hygiene, social distancing, wearing a mask, and vaccination. In addition, strengthening the level of research for the development of potential vaccines and anti-viral drugs, and a proper strategy to tackle the situation in case of new variants should be evaluated at the global level. It is also very important to monitor and expand genomic surveillance to keep track of the emergence of new variants, thus avoiding the spread of new diseases worldwide.

## Figures and Tables

**Figure 1 vaccines-10-01468-f001:**
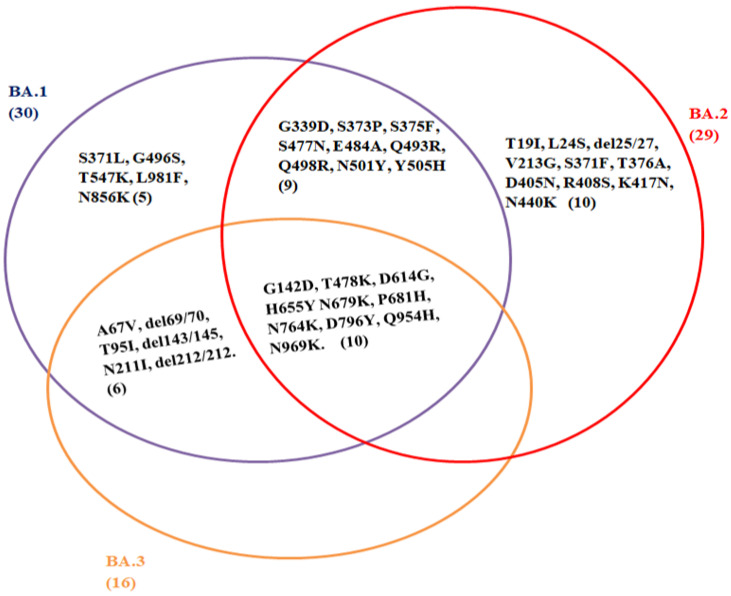
Comparison of spike protein mutations of Omicron sub-lineage: BA.1, BA.2, and BA.3. (GISAID: https://www.epicov.org/epi3/frontend#58aca) (Accessed on 10 February 2022). This Venn diagram represents the common mutation present among the sub-linages of Omicron variant; G142D, T478K, D614G, H655Y N679K, P681H, N764K, D796Y, Q954H, N969K. B.A.1 (30); A67V, del69/70, T95I, G142D, del143/145, N211I, del212/212, G339D, S371L, S373P, S375F, S477N, T478K, E484A, Q493R, G496S, Q498R, N501Y, Y505H, T547K, D614G, H655Y, N679K, P681H, N764K, D796Y, N856K, Q954H, N969K, L981F. B.A.2 (29); T19I, L24S, del25/27, G142D, V213G, G339D, S371F, S373P, S375F, T376A, D405N, R408S, K417N, N440K, S477N, T478K, E484A, Q493R, Q498R, N501Y, Y505H, D614G, H655Y, N679K, P681H, N764K, D796Y, Q954H, N969K. B.A.3 (16); A67V, del69/70, T95I, G142D, del143/145, N211I, del212/212, T478K, D614G, H655Y, N679K, P681H, N764K, D796Y, Q954H, N969K.

**Table 1 vaccines-10-01468-t001:** WHO-designated SARS-CoV-2 variants of concern (VOCs).

WHO Designation	Country First Origin	Pango Lineages	Variant Prevalence Countries as of 11 February 2022	GISAID	Next Strain	Mutation	Additional Amino Acid Changes Monitored
Alpha (18 December 2020)	United Kingdom, September-2020	9 Sub-lineages: B.1.1.7, Q.1, Q.4Q.5, Q.8, Q.7, Q.2, Q.6, Q.3.	United Kingdom (262,616)	GRY, GR/501Y.V1	20I/501Y.V1, 20B/501Y.V1	22 mutations (9 mutation spike protein, with deletion:del69/70, del144/144)	+S:484K +S:452R
Beta (18 December 2020)	South Africa, May-2020	B.1.351, B.1.351.3, B.1.351.2, B.1.351.5, B.1.351.1	South Africa (6885)	GH/501Y.V2	20H/501Y.V2	18 mutations (8 mutation at spike protein, with deletion:del241/243)	+S:L18F
Gamma (11 January 2021)	Brazil, November 2020.	23 Pango lineages currently associated with the Gamma variant.	Brazil (47,475)	GR/501Y.V3	20J/501Y.V3	23 mutations (12 mutation at spike protein)	+S:681H
Delta (11 May 2021)	India, Oct-2020	216 Pango lineages currently associated with the Delta variant	India (69,457)	G/452R.V3	21A/S:478K	29 mutations (8 mutation at spike protein, with deletion:del157/158)	+S:417N +S:484K
Omicron (26 November 2021)	South Africa, November 2021	BA.1, BA.1.1, BA.2BA.3	South Africa (4930)	GR/484A	21K, 21L, 21M.	∼50 mutations (30 mutations at spike protein, with deletion: del69/70, del143/145, and del212/212.)	+S:R346K

## Data Availability

Not applicable.

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
