# Peer review of "A Systematic Review on the Emergence of Omicron Variant and Recent Advancement in Therapies"

_vaccines, 2022, doi:10.3390/vaccines10091468_

Round 1

Reviewer 1 Report

This review of omicron variants from my perspective is divided into two parts: 

A rather interesting and useful second part reviewing the diagnosis and treatment options for these omicron variants.

A first part as a catch-all, giving the impression of an accumulation of raw data, with no real structure, no interpretation, no step back, not very helpful for understanding, source of confusion and contradiction.

Specific comments

Line 50 : “As per WHO, this novel variant also enhances immune or diagnostic evasion, disease severity, and significant transmissibility.”

Line 55   “early findings suggest less hospitalization for Omicron-positive patients as compared to the 56 Delta variant.

Line83 “However, it is less severe than 83 the previous variant …”

There seems to be an apparent contradiction between what is written in line 50, (enhance disease severity), and line 55 (less hospitalization) and 83 (less severe)

The emergence of omicron (B.1.1.529) and its sub-lineage

Line82 “This could be due to the high rate of mutation and replication in the Omicron variant as compared to other variants of concern…”

From my point of view, there is no evidence that this variant has a different mutation rate from the other variants (for me, the mutation rate refers to the error rate of the polymerase). It is a pity that the two hypotheses to explain this large number of mutations in this omicron variant, particularly at the level of the spike protein, are not discussed here, namely a prolonged infection in an immunosuppressed patient or possibly a passage in an animal host (with a clear preference for the first hypothesis). At the very least this part is clumsy.

Line 84 “After the announcement of Omicron by  WHO, three days later on 29 November 2021, this novel variant was detected in Australia, Belgium, Canada, Czech Republic, Denmark, France, Germany, Italy, Netherlands, and  the United Kingdom [11].” 

It is a particular way of saying things, without filter, without distance, without interpretation,

Line 92 « African pollution… » meaning population ?

Line 118. “It possesses immune-escaped properties among vaccinated individuals; however, it does not increase its transmissibility to vaccinated individuals with break- through infections.”. I'm not sure I understand the meaning of this sentence

This whole chapter is a compilation of facts delivered in a more or less crude way without any key of interpretation sometimes making little sense.

Mechanism of Omicron Variant ? 

I do not understand the meaning of this title in reference to the following paragraph.

Line 129 « 15 modifications is observed in a Ribosomal Binding Domain (RBD)…” Sincerely I am lost, what are we talking about? NSP1? from my point of view it creates a lot of confusion to use the same abbreviation "RBD" indifferently for Receptor Binding Domain which makes sense and for Ribosomal Binding Domain which induces a lot of confusion without really knowing what we are talking about and what are the consequences. 

Line 134 « mutation in Ribosomal Binding Domain of SARS-CoV-2 Omicron variant resulted in higher binding  affinity to human ACE2 protein” . now I'm completely lost, it doesn't make sense to me at all

Line 158 “which 15 modification is located at the ribosomal Binding Domain (RBD) region [31, 32]. 

two references in support of this sentence, neither of which mentions this Ribosomal Binding Domain. This notion comes back several times and I don't know what we are talking about

Line 227 “Deerain et al. [51] evaluated ten commercially available rapid antigen test kits to check their diagnostic performance (sensitivity) among the Delta and Omicron variants.” 

It would be interesting to know which Ag are detected in these different tests.

Line 326 « All these drugs paved as new hopes to fight against the COVID-19 pandemic.” Je reste surprise que l’hydroxychloroquine puisse être considérée comme un nouvel espoir de traitement…

Discussion, conclusion and prospect.

Line 410 : “The recent advancement in Anti-viral drugs may rise as a new hope to stop the spread of this variant.” Very optimistic and unlikely

Indeed 6 lines later: “This article has a limitation, as it lacks clinical data on the efficacy of mAbs and anti-viral drugs for the treatment of the patient with Omicron variant.” At least it is said

Line 423 “It has been almost 2 years since the emergence of this COVID pandemic.” it is more than two years

Line 431 “The emergence of mutation could be due to inequitable access to vaccines…”. non la pression de selection pour l’émergence de mutation est la transmissibilité et l’échappement à l’immunité de la population pas le taux de vaccination qui ne peut participer qu’à la seconde composante de cette pression de sélection. » No, the selection pressure for mutation emergence is the transmissibility and escape from population immunity, not the vaccination rate which can only participate in the second component of this selection pressure. At the very least it introduces confusion

Author Response

Comments and Suggestions for Authors

This review of omicron variants from my perspective is divided into two parts: 

A rather interesting and useful second part reviewing the diagnosis and treatment options for these omicron variants.

A first part as a catch-all, giving the impression of an accumulation of raw data, with no real structure, no interpretation, no step back, not very helpful for understanding, source of confusion and contradiction.

Specific comments

Line 50 : “As per WHO, this novel variant also enhances immune or diagnostic evasion, lower disease severity, and significant transmissibility.”

Line 55   “early findings suggest less hospitalization for Omicron-positive patients as compared to the 56 Delta variant.

Line83 “However, it is less severe than 83 the previous variant …”

There seems to be an apparent contradiction between what is written in line 50, (enhance disease severity), and line 55 (less hospitalization) and 83 (less severe)

 =less severe disease

The emergence of omicron (B.1.1.529) and its sub-lineage

Line82 “This could be due to the high rate of mutation and replication in the Omicron variant as compared to other variants of concern…”

From my point of view, there is no evidence that this variant has a different mutation rate from the other variants (for me, the mutation rate refers to the error rate of the polymerase). It is a pity that the two hypotheses to explain this large number of mutations in this omicron variant, particularly at the level of the spike protein, are not discussed here, namely a prolonged infection in an immunosuppressed patient or possibly a passage in an animal host (with a clear preference for the first hypothesis). At the very least this part is clumsy.

= This could be due to large number of mutation at the spike protein in the omicron variant.

Line 84 “After the announcement of Omicron by  WHO, three days later on 29 November 2021, this novel variant was detected in Australia, Belgium, Canada, Czech Republic, Denmark, France, Germany, Italy, Netherlands, and  the United Kingdom [11].” 

It is a particular way of saying things, without filter, without distance, without interpretation,

= Three days after the announcement of Omicron in 29 Nov 2021 by WHO, this novel variant was detected in Australia, Belgium, Canada, Czech Republic, Denmark, France, Germany, Italy, Netherlands, and the United Kingdom.

( https://www.ncbi.nlm.nih.gov/pmc/articles/PMC8634699/)

Line 92 « African pollution… » meaning population ?

= Population.

Line 118. “It possesses immune-escaped properties among vaccinated individuals; however, it does not increase its transmissibility to vaccinated individuals with break- through infections.”. I'm not sure I understand the meaning of this sentence [23]

=BA.2 variant enhanced immune evasion as such it is was found to be equally infectious and transmissible among the population despite of being vaccinated.

(https://doi.org/10.1101/2022.01.28.22270044 )

This whole chapter is a compilation of facts delivered in a more or less crude way without any key of interpretation sometimes making little sense.

Mechanism of Omicron Variant ? 

=Mechanism of invasion.

I do not understand the meaning of this title in reference to the following paragraph.

Line 129 « 15 modifications is observed in a Receptor  Binding Domain (RBD)…” Sincerely I am lost, what are we talking about? NSP1? from my point of view it creates a lot of confusion to use the same abbreviation "RBD" indifferently for Receptor Binding Domain which makes sense and for Ribosomal Binding Domain which induces a lot of confusion without really knowing what we are talking about and what are the consequences. 

Line 134 « mutation in Receptor  Binding Domain of SARS-CoV-2 Omicron variant resulted in higher binding  affinity to human ACE2 protein” . now I'm completely lost, it doesn't make sense to me at all

Line 158 “which 15 modification is located at the Receptor Binding Domain (RBD) region [31, 32]. 

two references in support of this sentence, neither of which mentions this Ribosomal Binding Domain. This notion comes back several times and I don't know what we are talking about

Line 227 “Deerain et al. [51] evaluated ten commercially available rapid antigen test kits to check their diagnostic performance (sensitivity) among the Delta and Omicron variants.” 

It would be interesting to know which Ag are detected in these different tests.

= to detect nucleocapsid (N) protein.

Line 326 « All these drugs paved as new hopes to fight against the COVID-19 pandemic.” Je reste surprise que l’hydroxychloroquine puisse être considérée comme un nouvel espoir de traitement…

=updated

Discussion, conclusion and prospect.

Line 410 : “The recent advancement in Anti-viral drugs may rise as a new hope to stop the spread of this variant.” Very optimistic and unlikely

Indeed 6 lines later: “This article has a limitation, as it lacks clinical data on the efficacy of mAbs and anti-viral drugs for the treatment of the patient with Omicron variant.” At least it is said

Line 423 “It has been almost 2 years since the emergence of this COVID pandemic.” it is more than two years

 =revised.

Line 431 “The emergence of mutation could be due to inequitable access to vaccines…”. non la pression de selection pour l’émergence de mutation est la transmissibilité et l’échappement à l’immunité de la population pas le taux de vaccination qui ne peut participer qu’à la seconde composante de cette pression de sélection. » No, the selection pressure for mutation emergence is the transmissibility and escape from population immunity, not the vaccination rate which can only participate in the second component of this selection pressure. At the very least it introduces confusion

=revised

Reviewer 2 Report

This manuscript aims for a comprehensive review of anti-COVID-19 therapeutics in the context of the dominant Omicron SARS-CoV-2 variant. The manuscript is in some parts lossy written, and the use of English is deficient. An edited file with the clearest spelling errors and incorrect use of language has been attached to the comments, hopping it helps the writers. Each section of the manuscript seems to have been assigned to a different author, but nobody took the time to carefully read and correct the final version. There are also very fundamental errors, like for example the use of incorrect use of Ribosoma Binding Domain instead of Receptor Binding Domain term (RBD). Relevant literature is also missing. For example, the experimental affinity of Omicron RBD towards human ACE2 was already published in Nat. Comm. at the time the manuscript was written but is not cited (see https://doi.org/10.1038/s41392-021-00863-2). Instead, they cite a manuscript which to-date has not been peer-reviewed and shows opposite results solely based on computational simulations, completely ignoring experimental data (see doi: 10.1101/2022.01.24.477633). This lack of knowledge leads to biased conclusions.

In addition, the list of therapeutics authorized against COVID-19 is incomplete. To give an example, Baricitinib (approved for emergency use 11/19/2020) is not included. I do not understand why the FDA page of emergency use authorization for COVID-19 has not been listed as a source in Table 2. (https://www.fda.gov/emergency-preparedness-and-response/mcm-legal-regulatory-and-policy-framework/emergency-use-authorization#coviddrugs).

In conclusion, I cannot recommend the acceptance of this manuscript for its publication in Vaccines. 

Author Response

This manuscript aims for a comprehensive review of anti-COVID-19 therapeutics in the context of the dominant Omicron SARS-CoV-2 variant. The manuscript is in some parts lossy written, and the use of English is deficient. An edited file with the clearest spelling errors and incorrect use of language has been attached to the comments, hopping it helps the writers. Each section of the manuscript seems to have been assigned to a different author, but nobody took the time to carefully read and correct the final version. There are also very fundamental errors, like for example the use of incorrect use of Ribosoma Binding Domain instead of Receptor Binding Domain term (RBD). Relevant literature is also missing. For example, the experimental affinity of Omicron RBD towards human ACE2 was already published in Nat. Comm. at the time the manuscript was written but is not cited (see https://doi.org/10.1038/s41392-021-00863-2). Instead, they cite a manuscript which to-date has not been peer-reviewed and shows opposite results solely based on computational simulations, completely ignoring experimental data (see doi: 10.1101/2022.01.24.477633). This lack of knowledge leads to biased conclusions.

Lupala, C.S.; Ye, Y.; Chen, H.; Su, XD.; Liu, H. Mutations on RBD of SARS-CoV-2 Omicron variant result in stronger binding to human ACE2 receptor. BiochemBiophys Res Commun 2022, 590, 34-41. doi:10.1016/j.bbrc.2021.12.079

Wu, L.; Zhou, L.; Mo, M.; Liu, T.; Wu, C.; Gong, C.; Lu, K.; Gong, L.; Zhu, W.; Xu, Z. SARS-CoV-2 Omicron RBD shows weaker binding affinity than the currently dominant Delta variant to human ACE2. Signal transduction and targeted therapy, 20227, 8. https://doi.org/10.1038/s41392-021-00863-2

In addition, the list of therapeutics authorized against COVID-19 is incomplete. To give an example, Baricitinib (approved for emergency use 11/19/2020) is not included. I do not understand why the FDA page of emergency use authorization for COVID-19 has not been listed as a source in Table 2. (https://www.fda.gov/emergency-preparedness-and-response/mcm-legal-regulatory-and-policy-framework/emergency-use-authorization#coviddrugs).

It has been updated.

Round 2

Reviewer 1 Report

This new version is significantly improved, easier to read, more understandable and overall useful and interesting

Author Response

Response: Thank you so much for your encouraging words.

Reviewer 2 Report

The authors have substantially improved the use of language and update references and data listed in tables, which enhances the overall quality of the manuscript. The authors also corrected the journal of the article from Wu. L. et al.  I cited in my last review from Nat. Comm. (incorrect) into Signal transduction and targeted therapy, which I appreciate since this is the correct one. 

There are still some minor details like the use of acronyms, where the author introduce some of them several time throughout the manuscript, when only the first one suffices. For example, RBD is explained 7 times, one is even wrong (line 1207). The same applies for VOC, which is explained 9 times, and for VOI, which is introduced 2 times.

Once corrected, I think that the manuscript meets the requirements for its publication in Vaccines.

Author Response

The authors have substantially improved the use of language and update references and data listed in tables, which enhances the overall quality of the manuscript. The authors also corrected the journal of the article from Wu. L. et al.  I cited in my last review from Nat. Comm. (incorrect) into Signal transduction and targeted therapy, which I appreciate since this is the correct one. 

There are still some minor details like the use of acronyms, where the author introduce some of them several time throughout the manuscript, when only the first one suffices. For example, RBD is explained 7 times, one is even wrong (line 1207). The same applies for VOC, which is explained 9 times, and for VOI, which is introduced 2 times.

Response: Thank you so much for your valuable comments. We have corrected the minor details in the revised version of the manuscript.